# The Efficacy and Safety Herbal Medicine for Symptom Management After HIFU Treatment in Adenomyosis: A Systematic Review and Meta-Analysis

**DOI:** 10.3390/ph18060843

**Published:** 2025-06-04

**Authors:** Eun-Jin Kim, Young-Shin Shim, Hyun-Kyung Sung, Sang-Yeon Min

**Affiliations:** 1Department of Pediatrics of Korean Medicine, Korean Medicine Hospital, Dongguk University Bundang Medical Center, Seongnam-si 13601, Gyeonggi-do, Republic of Korea; utopialimpid@naver.com; 2Department of Pediatrics of Korean Medicine, Graduate School of Dongguk University, Seoul 04620, Republic of Korea; simyoungshin@naver.com; 3Department of Pediatrics of Korean Medicine, Korean Medicine Hospital, Dongguk University Ilsan Medical Center, Goyang-si 10326, Gyeonggi-do, Republic of Korea; 4Department of Education, College of Korean Medicine, Dongguk University, Gyeongju-si 38066, Gyeongsangbuk-do, Republic of Korea; shksolar@gmail.com

**Keywords:** adenomyosis, herbal medicine, high-intensity focused ultrasound, systematic reviews, meta-analysis

## Abstract

**Background/Objectives:** Adenomyosis (AM) is a hormone-dependent gynecological disorder that negatively impacts the quality of life and fertility of reproductive-age women. This study aimed to evaluate the effectiveness of herbal medicine (HM) as a post-treatment strategy following high-intensity focused ultrasound (HIFU) therapy. **Methods**: English, Chinese, and Korean databases were systematically searched up to 24 March 2025. Eligible randomized controlled trials (RCTs) compared HM administration after HIFU therapy with HIFU therapy alone. Statistical analyses included mean difference (MD), standardized mean difference (SMD), and risk ratio (RR) with 95% confidence intervals (CIs). Evidence quality was assessed using GRADE approach. The protocol was registered with INPLASY (No.: INPLASY202530088). **Results**: Fourteen RCTs involving 1259 patients were included in the review. HM administration after HIFU therapy showed superior efficacy over HIFU therapy alone in reducing uterine volume (MD = −11.84, 95% CI: −13.74 to −9.95; *p* < 0.00001), adenomyotic lesion volume (MD = −2.86, 95% CI: −3.29 to −2.43; *p* < 0.00001), serum CA125 levels (SMD = −1.49, 95% CI: −2.41 to −0.58; *p* < 0.00001), serum estradiol (E2) levels (SMD = −1.28, 95% CI: −1.54 to −1.03; *p* < 0.0001), and improvements in dysmenorrhea (MD = −0.54, 95% CI: −1.06 to −0.02; *p* < 0.00001) **Conclusions**: HM may be an effective and safe adjunct to HIFU for managing AM. However, further high-quality RCTs with long-term follow-up are needed to validate these findings.

## 1. Introduction

Adenomyosis (AM) is a hormone-dependent gynecological condition characterized by the ectopic presence of endometrial tissue within the myometrium, leading to uterine enlargement and a range of debilitating symptoms [1,2] (Figure 1). It predominantly affects reproductive-age women and manifests clinically as abnormal uterine bleeding (such as menorrhagia or intermenstrual spotting), chronic pelvic pain (including dysmenorrhea and dyspareunia), reduced fertility, or an increased risk of miscarriage, collectively contributing to significant impairments in the quality of life [3,4].

Recent data suggest an increasing incidence of AM, with a notable shift toward a younger age at diagnosis. Among symptomatic individuals, the estimated prevalence is approximately 30–35%, and infertility has been reported in nearly one-fifth of affected women [5,6]. Although total hysterectomy is currently considered the definitive treatment, it is unsuitable for patients wishing to preserve their reproductive potential or maintain their uteri [7]. As an alternative, high-intensity focused ultrasound (HIFU) therapy has emerged as a promising noninvasive uterus-conserving approach. The use of precisely targeted thermal energy under real-time guidance via magnetic resonance (MRgHIFU) or ultrasound (USgHIFU) allows for the ablation of adenomyotic tissue without surgical incision [8]. Numerous studies have demonstrated that HIFU therapy significantly reduces menstrual volume and improves pain symptoms associated with AM [9]. Nonetheless, its ability to eliminate lesions is limited, and recurrence remains a clinical concern. The likelihood of recurrence may be influenced by individual factors such as uterine size, lesion characteristics (e.g., diffuse type), and degree of ablation achieved [8]. Additionally, although HIFU therapy is regarded as less detrimental to future fertility than invasive surgical options, evidence on post-treatment reproductive outcomes remains limited [10], and standardized post-HIFU management protocols are currently lacking.

In clinical practice, herbal medicine (HM) is commonly administered along with HIFU therapy to enhance therapeutic outcomes and mitigate symptom recurrence in patients with AM. A previous systematic review and meta-analysis [11] suggested the potential synergistic benefits of combining HM with HIFU therapy versus HIFU therapy alone. However, the review included gray literature and a limited number of randomized controlled trials (RCTs), which may have introduced bias and reduced the overall strength of the evidence. Given these developments, the present study aimed to provide an updated and methodologically rigorous systematic review and meta-analysis to evaluate the efficacy and safety of HM as an adjunct to HIFU therapy for the treatment of AM.

## 2. Methods

### 2.1. Protocol and Registration

This systematic review and meta-analysis were conducted in accordance with the Preferred Reporting Items for Systematic Reviews and Meta-Analyses (PRISMA) guidelines [12] (see Appendix A). The review protocol was prospectively registered in INPLASY (registration no. INPLASY202530088) on 21 March 2025, and is publicly available at https://inplasy.com/inplasy-2025-3-008/ (accessed on 1 June 2025).

### 2.2. Eligibility Criteria

#### 2.2.1. Types of Studies

This review included RCTs that evaluated the therapeutic effects of HM administered after HIFU therapy on AM. Non-RCTs, RCT protocols, animal experiments, case reports, theses, surveys, and review articles were excluded from this review.

#### 2.2.2. Inclusion and Exclusion Criteria

The participants were patients diagnosed with AM according to the diagnostic criteria outlined in the Obstetrics and Gynecology section.

Exclusion criteria included contraindications to ultrasound ablation, the need for surgical intervention, the presence of severe gynecological conditions (e.g., malignant tumors), significant hepatic or renal impairment, cardiovascular disease, or other serious systemic disorders. Pregnant or lactating women were also excluded.

#### 2.2.3. Types of Interventions

The experimental group received HIFU therapy followed by HM treatment, with no restrictions on the specific formulation (e.g., oral administration, external application, or enema).

#### 2.2.4. Types of Comparisons

The control group received only HIFU therapy. Conventional treatment was permitted if it was administered equally in both groups.

#### 2.2.5. Types of Outcome Measures

The primary outcomes were the uterine volume or adenomyotic lesion volume measured by transvaginal ultrasound, serum CA125 and E2 levels in the peripheral blood, and severity of dysmenorrhea assessed using the visual analog scale (VAS). Additional outcomes included the total effective rate (TER), menstrual bleeding volume evaluated using the pictorial blood loss assessment chart (PBAC), and incidence of adverse events.

### 2.3. Data Sources and Search Strategy

A comprehensive literature search was conducted across 11 electronic databases up to 24 March 2025, without restrictions on the language or year of publication. The databases included three English language databases (MEDLINE via PubMed, Excerpta Medica dataBASE [EMBASE], and the Cochrane Central Register of Controlled Trials [CENTRAL]); three Chinese databases (China National Knowledge Infrastructure [CNKI], Chinese Scientific Journal Database [VIP], and WanFang Data); and five Korean medical databases (Oriental Medicine Advanced Searching Integrated System [OASIS], Korean Studies Information Service System [KISS], Korea Citation Index [KCI], Research Information Sharing Service [RISS], and the Korean Medical Database [KMbase]). The search strategy included the terms “adenomyosis” and “herbal medicine”, which were adapted to meet the linguistic and indexing conventions of each database. Detailed search strategies and results for each database are provided in Appendix A.

### 2.4. Study Selection and Data Extraction

#### 2.4.1. Study Selection

Two authors (E.-J.K. and Y.S.S.) independently screened the titles and abstracts of all retrieved records to identify potentially eligible studies. Articles deemed relevant were then subjected to a full-text review, which was conducted based on the predefined inclusion and exclusion criteria to determine final eligibility. Discrepancies between the two authors were resolved through discussion, and if a consensus could not be reached, a third author (S.Y.M.) was consulted to arbitrate and finalize the decision.

#### 2.4.2. Data Extraction

Two authors (E.-J.K. and Y.S.S.) independently extracted data from the included studies. Any discrepancies were resolved through discussion among the authors. In case of missing or unclear information, the corresponding authors were contacted via email. Extracted data included the first author’s name, year of publication, sample size, total duration of treatment, participant characteristics, treatment interventions and comparators, outcome measures, adverse events, and information relevant to risk of bias (RoB) assessment.

### 2.5. RoB Assessment

Two independent authors (E.J.K. and Y.S.S.) evaluated the risk of bias in the included studies using the Cochrane RoB 2 tool, following the guidelines of the Cochrane Handbook for Systematic Reviews of Interventions [13]. Each study was rated as having a low, some concerns, or a high risk of bias. The assessment addressed five key domains: randomization process, deviations from the intended interventions, missing outcome data, measurement of outcomes, and selection of reported results. Discrepancies between the reviewers were resolved through consensus following discussions with all authors.

### 2.6. Statistical Analysis

All included studies were initially subjected to qualitative synthesis. When two or more studies reported comparable continuous or dichotomous outcome measures, a meta-analysis was conducted using Review Manager (RevMan) software (version 5.4; Cochrane Collaboration, London, UK). For dichotomous variables, risk ratios (RRs) with 95% confidence intervals (CIs) were calculated, whereas continuous variables were analyzed using either mean differences (MDs) or standardized mean differences (SMDs), depending on the consistency of the measurement units, each with corresponding 95% CIs.

#### 2.6.1. Heterogeneity Assessment

Heterogeneity among the included studies was assessed using Higgins I^2^ statistic [14]. An I^2^ value of ≥50% was considered to indicate substantial heterogeneity, in which case a random-effects model was applied for data synthesis. Conversely, an I^2^ value of <50% indicated low heterogeneity, and a fixed-effects model was employed.

#### 2.6.2. Reporting Bias Assessment

Publication bias was evaluated for outcomes that included at least ten studies. Funnel plots were examined for visual asymmetry. Where asymmetry suggested a potential bias, further statistical assessments were performed. These included Egger’s regression test, Rosenthal’s fail-safe N, and the trim-and-fill method using R Studio (version 4.5.0; RStudio, PBC, Boston, MA, USA) utilizing the “meta” package with default settings.

#### 2.6.3. Subgroup and Sensitivity Analyses

If significant heterogeneity was observed in the meta-analysis, subgroup analyses were performed to explore the potential sources of variability. Subgroup analyses were also conducted based on the presence or absence of concurrent conventional treatment, provided that sufficient data were available. In addition, sensitivity analyses were conducted independently for each outcome when more than ten studies were included.

### 2.7. Quality of Evidence

The certainty of evidence was assessed using the Grading of Recommendations Assessment, Development, and Evaluation (GRADE) approach, following the standardized criteria outlined at http://gradepro.org (accessed on 4 May 2025). Evidence quality was evaluated across five domains: risk of bias, inconsistency, indirectness, imprecision, and publication bias. Based on these assessments, the overall certainty of evidence for each outcome was rated as high, moderate, low, or very low according to the GRADE framework.

## 3. Results

### 3.1. Results of Literature Search

After applying the search strategy across all databases, 2787 records were identified, including 133 from English-language databases, 2632 from Chinese databases, and 19 from Korean databases. After removing duplicates, 2536 records remained. Based on the title and abstract screening, 129 studies were selected for full-text review. However, 100 non-RCT studies, 11 studies involving inappropriate interventions, and 4 studies not relevant to AM were excluded. Ultimately, 14 RCTs [15,16,17,18,19,20,21,22,23,24,25,26,27,28] met the inclusion criteria and were included in this systematic review and meta-analysis (Figure 2).

### 3.2. Study Characteristics

All 14, included RCTs, were conducted in China. The studies were published between 2017 and 2025 with sample sizes ranging from 60 to 130 participants. The treatment duration varied from 7 days to 6 months. Participants’ ages ranged from 21 to 53 years. The duration of illness also varied, with the shortest reported as 2.16 ± 0.49 years and the longest as 6.02 ± 1.98 years (mean ± SD) (Table 1).

Four studies [16,21,24,27] employed diagnostic criteria from the Obstetrics and Gynecology, 9th edition [29], whereas five studies [17,20,23,24,26] used diagnostic criteria from the Chinese Guidelines for the Diagnosis and Treatment of Endometriosis, 3rd edition [30]. Several other studies used alternative diagnostic criteria [15,16,18,19,21,28] or authors’ definitions [22,25]. Notably, five studies applied multiple diagnostic criteria to confirm the diagnosis: one study [15] integrated three diagnostic criteria, and four studies [16,18,21,24] combined two criteria.

### 3.3. Interventions

Of the included studies, ten studies [15,17,18,19,20,21,23,26,27,28] administered HM orally, one study [22] utilized external application, one study [16] employed both oral decoction and external application, and two studies [24,25] administered HM via enema. The formulations used included decoctions [15,16,17,20,21,23,26,28], pills [18,19], and granules [27]. The detailed compositions, dosages, and administration frequencies of the HM treatments are summarized in Table 2. *Angelicae gigantis radix* was the most frequently used herb, appearing in 9 studies, followed by *Sparganii rhizoma*, *Curcumae rhizoma*, *Paeoniae radix*, *Cnidii rhizoma*, and *Corydalis tuber* each appearing six times. *Poria sclerotium*, *Myrrha*, *Cinnamomi ramulus*, and *Paeoniae radix alba* appeared five times (Appendix A). Two studies combined HM with acupuncture: one study [17] used acupuncture alone, whereas the other [21] employed both acupuncture and auricular acupuncture. HIFU therapy was administered in all included studies. Two studies [24,26] incorporated gonadotropin-releasing hormone agonist (GnRHa) therapy after HIFU therapy, whereas one study [19] utilized the levonorgestrel-releasing intrauterine system (LNG-IUS) as a post-HIFU intervention in both groups. Detailed information regarding the HIFU procedures, including the methods and parameters used across the studies, is presented in Appendix A.

### 3.4. Outcome Measures

The primary outcome measures were the uterine volume, adenomyotic lesion volume, serum CA125 and E2 levels, and dysmenorrhea severity measured using the VAS. Uterine volume was assessed in seven studies [17,18,19,20,23,24,28], among which three studies were excluded from the meta-analysis because of unit inconsistencies [23], implausible post-treatment values [17], and missing outcome data [20]. Consequently, four studies [18,19,24,28] were included in the final analysis. All five studies reporting pre- and post-treatment changes in adenomyotic lesion volume [15,19,24,25,28] were included in the meta-analysis. Serum CA125 levels were measured in seven studies [15,18,20,23,26,27,28], whereas serum E2 levels were evaluated in three studies [19,27,28]. VAS scores for dysmenorrhea were reported in 11 studies [15,16,17,18,19,20,22,23,24,25,27]. Most studies employed a standard 0–10 VAS to assess the severity of menstrual pain before and after treatment. One study [16] was excluded from the meta-analysis because it used a modified VAS. Thus, a total of 10 studies [15,17,18,19,20,22,23,24,25,27] were included in the final meta-analysis for dysmenorrhea.

The secondary outcome measures were the TER, menstrual bleeding volume assessed using PBAC, and incidence of adverse events. TER was reported in eight studies [17,19,20,22,24,26,27,28]. PBAC scores for menstrual volume were reported in eight studies [15,16,18,19,20,22,23,24]. Among these, two studies that employed unspecified assessment methods [19,20] and two studies [15,16] that utilized modified PBAC scoring systems were excluded. Consequently, four studies [18,22,23,24] were included in the final analysis. Adverse events were documented in eight studies [15,18,19,21,23,24,25,27]. One study [19] was excluded from the meta-analysis because it did not report the incidence of adverse events. Consequently, seven studies [15,18,21,23,24,25,27] were included in the final meta-analysis (Appendix A). Comprehensive details of all secondary outcome measures and their corresponding *p*-values are provided in Appendix A.

### 3.5. Quality Assessment

Risk of bias was assessed using the Cochrane RoB 2 tool. Most of the included studies were evaluated as having concerns in the randomization process domain, primarily because of insufficient information regarding allocation sequence concealment. One study [22] was judged to have a high risk of bias as participants were assigned to intervention or control groups based on the order of enrollment, which did not ensure adequate allocation concealment. There were no deviations from the intended interventions, indicating that the planned treatments or procedures were consistently implemented for all participants. Therefore, all included studies were judged to have a low risk of bias in the domain of deviation from the intended interventions. Most of the studies had a low risk of bias in the missing outcome data domain. However, one study [20] was rated otherwise, as it failed to report outcomes that had been initially pre-specified as primary or secondary endpoints. Similarly, most studies had a low risk of bias in the outcome measurement domain. However, two studies [17,23] raised concerns due to inconsistencies in post-treatment values and inaccuracies in the reported measurement units. In terms of the selection of the reported results, all studies were assessed to have a low risk of bias.

Consequently, with four studies [17,20,22,23] categorized as having a high risk of bias, the remaining studies were judged to have concerns regarding the overall risk of bias. A summary of these assessments is provided in Figure 3 and Figure 4.

### 3.6. Meta-Analysis

A meta-analysis was conducted to compare the efficacy of HM administration after HIFU therapy with that of HIFU therapy alone. When significant heterogeneity was detected, a subgroup analysis was performed with or without concurrent conventional treatment.

#### 3.6.1. Uterine and Lesion Volumes

Four studies [18,19,24,28] involving 304 patients that reported uterine volumes before and after treatment were included in the meta-analysis. The pooled analysis demonstrated an MD of −11.84 (95% CI: −13.74 to −9.95), indicating a statistically significant reduction in uterine volume following the intervention. Heterogeneity among the studies was low (I^2^ = 24%, *p* = 0.27), suggesting consistent findings across trials; therefore, a fixed-effects model was applied. The overall effect was highly significant (Z = 12.27, *p* < 0.00001), supporting the robust efficacy of the treatment in reducing the uterine size (Figure 5).

Similarly, five studies [15,19,24,25,28] involving 420 patients that reported adenomyotic lesion volumes before and after treatment were included in the meta-analysis. The analysis revealed an MD of −2.86 (95% CI: −3.29 to −2.43), indicating a significant reduction in lesion volume after treatment. Moderate heterogeneity was observed (I^2^ = 42%, *p* = 0.14); however, this was not statistically significant, indicating an acceptable level of consistency across studies. Therefore, we used a fixed effects model in this study. The test for the overall effect showed a highly significant result (Z = 13.09, *p* < 0.00001), further confirming the effectiveness of the intervention in reducing adenomyotic lesions (Figure 6).

#### 3.6.2. Serum Biomarkers

The SMD approach was adopted across studies to enhance analytical consistency and comparability and to address methodological and clinical differences in the assessment of CA125 and E2 outcomes, such as variations in baseline concentrations, measurement time points, and laboratory assay techniques.

Seven studies [15,18,20,23,26,27,28] involving 616 patients reported pre- and post-treatment serum CA125 levels. The meta-analysis revealed a significant overall effect (SMD = −1.49; 95% CI: −2.41 to −0.58), indicating a statistically significant reduction in CA125 levels following the intervention. However, substantial heterogeneity was observed (I^2^ = 96%, *p* < 0.00001), reflecting considerable variability in treatment effects across studies. Therefore, a random-effects model was used. Despite this high level of heterogeneity, the overall effect remained statistically significant (Z = 3.22, *p* = 0.001), supporting the potential therapeutic efficacy of the intervention. Nonetheless, the variation among the studies warrants a cautious interpretation of the magnitude of the effects (Figure 7).

Three studies [19,27,28] enrolling 294 patients evaluated the changes in serum E2 levels before and after treatment. The meta-analysis revealed a significant pooled effect (SMD = −1.28; 95% CI: −1.54 to −1.03), indicating a clear and consistent reduction in E2 levels following the intervention. Importantly, no heterogeneity was detected across these studies (I^2^ = 0%, *p* = 0.70), supporting the use of a fixed-effects model. The overall effect was highly significant (Z = 9.98, *p* < 0.0001), suggesting robust and reliable evidence for the efficacy of the intervention in reducing serum E2 levels (Figure 8).

#### 3.6.3. VAS Score for Dysmenorrhea

As all included studies [15,17,18,19,20,22,23,24,25,27] assessed dysmenorrhea using 0–10 VAS, the MD approach was employed in the meta-analysis to facilitate clinically interpretable results. A meta-analysis of 10 studies, including 990 patients, evaluating changes in dysmenorrhea severity before and after treatment demonstrated an MD of −0.54 (95% CI: −1.06 to −0.02), indicating a statistically significant reduction in menstrual pain. However, substantial heterogeneity was detected (I^2^ = 96%, *p* < 0.00001), suggesting large variability in effect sizes across studies.

Subgroup analysis based on intervention type revealed that the overall treatment effect was statistically significant (Z = 2.03, *p* = 0.04; Figure 9). However, heterogeneity within the subgroups remained substantial (I^2^ = 96%, *p* < 0.00001), and a statistically significant difference between the subgroups was observed (I^2^ = 98.1%, *p* < 0.00001). This indicates that treatment responses differed significantly depending on the type of intervention and that considerable variability existed even within each subgroup. These findings suggest that while the treatment may be effective overall, the magnitude and consistency of its effects are influenced by the intervention applied.

#### 3.6.4. TER

A meta-analysis of eight RCTs [17,19,20,22,24,26,27,28] involving 764 participants evaluated the TER as the outcome. The pooled analysis showed a statistically significant benefit in favor of the treatment group (RR: 1.15, 95% CI: 1.08–1.22, *p* = 0.0001), indicating a higher clinical response rate compared with that in the control group. The between-study heterogeneity was low (I^2^ = 26%, *p* = 0.22), justifying the use of a fixed-effects model. The overall effect was statistically robust (Z = 4.60), supporting the efficacy of the intervention in improving the TER (Figure 10).

#### 3.6.5. PBAC Score for Menstrual Volume

A meta-analysis of four studies [18,22,23,24] comprising 350 participants evaluated menstrual volume using PBAC before and after treatment. The pooled results showed a significant MD of −8.75 (95% CI: −11.51 to −5.99), indicating a statistically significant reduction in menstrual volume following treatment (*p* < 0.00001). Heterogeneity among the studies was low (I^2^ = 14%, *p* = 0.32), justifying the use of a fixed-effects model. The overall effect was highly significant (Z = 6.21), suggesting that the intervention consistently and effectively reduced excessive menstrual bleeding, as measured by the PBAC (Figure 11).

#### 3.6.6. Adverse Events

A meta-analysis of eight studies [15,18,19,21,23,24,25,27], including a total of 717 patients, assessed the incidence of adverse events. The pooled analysis yielded a RR of 0.54 (95% CI: 0.40–0.72), indicating that patients in the treatment group had a significantly lower risk of experiencing adverse effects compared to those in the control group (*p* < 0.001). No heterogeneity was observed across the included studies (I^2^ = 0%, *p* = 0.72), sup-porting the application of a fixed-effects model. The test for the overall effect was highly significant (Z = 4.15), demonstrating that the intervention was associated with a favorable safety profile and reduced incidence of adverse events (Figure 12, Appendix A).

### 3.7. Publication Bias

The funnel plot of the VAS score for dysmenorrhea indicated a potential publication bias. Publication bias was not assessed for other outcome measures owing to the inclusion of <10 studies. Visual inspection of the funnel plot revealed mild asymmetry, suggesting a potential publication bias (Figure 13). However, Egger’s regression test showed no statistically significant asymmetry (intercept = −0.68, *p* = 0.7329), and the trim-and-fill method did not impute any additional studies. Furthermore, Rosenthal’s fail-safe N was calculated as 693, indicating that the overall results are unlikely to be overturned by unpublished studies with null effects. These findings suggest that the meta-analysis results were robust and minimally affected by publication bias.

### 3.8. Sensitivity Analyses

To evaluate the robustness of the pooled estimates, sensitivity analyses were performed by sequentially excluding individual studies on both serum CA125 levels and VAS scores for dysmenorrhea. For CA125, the overall treatment effect remained stable across all iterations (SMD range: −1.19 to −1.69), with statistical significance consistently maintained (*p* < 0.00001). Notably, heterogeneity remained high (I^2^ > 94%) throughout the study, suggesting that the observed variability could not be attributed to any single study (Appendix A). For VAS scores in dysmenorrhea, leave-one-out sensitivity analysis revealed a pooled MD ranging from −0.35 to −0.74, with all iterations consistently favoring the intervention group. Although the 95% confidence intervals of some iterations approached or slightly included the null value, the statistical significance was preserved in all cases (*p* < 0.00001). Heterogeneity remained substantial (I^2^ = 94–97%) across iterations, indicating that the variability may stem from clinical or methodological differences rather than the influence of a single outlier (Appendix A).

### 3.9. GRADE Certainty of Evidence

The certainty of the evidence was evaluated using the GRADE approach, and the findings are summarized in Table 3. The VAS scores for dysmenorrhea, TER, and adverse events were rated with moderate certainty. In contrast, the certainty of evidence for uterine volume, adenomyotic lesion volume, serum CA125 and E2 levels, and PBAC scores for menstrual volume was assessed as low. The detailed justifications for downgrading the quality of evidence for each outcome domain are provided in Table 3.

## 4. Discussion

### 4.1. Summary of This Review

This review aimed to evaluate the efficacy and safety of HM as a posttreatment intervention following HIFU therapy in patients with AM. After a comprehensive literature search, 14 RCTs involving 1259 participants were included in the meta-analysis. The combination of HM and HIFU therapy demonstrated significant therapeutic benefits compared with HIFU therapy alone. Notable improvements were observed in the reduction in uterine volume, adenomyotic lesion volume, and severity of dysmenorrhea as well as a significant decrease in menstrual bleeding. Serum CA125 and E2 levels were also significantly reduced. Moreover, HM was associated with a lower incidence of post-HIFU adverse effects such as pelvic pain and abnormal vaginal bleeding. These findings suggest that HM may serve as a valuable adjunctive strategy for postprocedural management and recurrence prevention in patients with AM.

### 4.2. Clinical Implication, Limitations and Suggestions

The treatment strategies for AM vary depending on the patient’s age and desire for future fertility. While total hysterectomy or adenomyomectomy are considered definitive treatments, hormonal suppression using GnRHa or contraceptive methods such as LNG-IUS is commonly used to manage symptoms in clinical settings [31]. For patients who wish to preserve their uterus, particularly reproductive-age women planning pregnancy, nonsurgical options such as HIFU therapy are often preferred. However, as AM tends to recur, continuous post-treatment management is essential. In this context, HM offers a meaningful complementary approach.

One included study [18] reported pregnancy rates, showing that the experimental group receiving HM had a significantly higher pregnancy rate (36.67%) at six months post-treatment than the control group (13.33%, *p* < 0.05). For reproductive-age women, preserving fertility is as crucial as relieving symptoms. The combination of HIFU therapy and HM may be a promising treatment strategy. Nevertheless, further research is required to confirm and improve its impact on pregnancy outcomes.

The pharmacological effects of commonly used HM ingredients are summarized as follows: Coniferyl ferulate, a bioactive compound found in *Angelicae gigantis radix*, has been reported to enhance antioxidant defense and promote hematopoiesis through activation of the JAK2/STAT5 signaling pathway [32,33]. Sparstolonin B, a novel bioactive compound isolated from *Sparganii rhizoma*, significantly inhibits neovascularization by halting endothelial cell cycle progression through G_1_/S checkpoint arrest. This suppression of microvessel formation in adenomyotic lesions limits oxygen and nutrient supply to ectopic endometrial tissue, thereby restricting its expansion [34]. Curcumin, a polyphenolic compound derived from *Curcumae rhizoma*, has shown hormone-modulating activity by reducing estrogen synthesis and downregulating estrogen-dependent cellular proliferation. These effects help to suppress the hormone-driven progression of adenomyotic tissue [35]. Paeoniflorin, a bioactive compound found in *Paeoniae radix* has been shown to reduce inflammation and alleviate pain by suppressing proinflammatory cytokines and COX-2, while enhancing anti-inflammatory responses [21,36]. Ligustilide, a major bioactive compound in *Cnidii rhizoma*, has been shown to reduce oxidative stress by inhibiting reactive oxygen species (ROS) and modulating the NF-κB signaling pathway [37]. Quercetin, a bioactive compound found in *Corydalis tuber*, possesses analgesic properties and regulates inflammatory mediators such as TNF, IL-6, and COX-2. It also inhibits VEGF-related angiogenesis, which is associated with the mechanisms of neuropathic pain [38].

This review included studies that utilized not only oral HM but also external and enema-based applications, highlighting the potential therapeutic versatility of HM in various formulations. In practice, the external application of HM may provide faster local effects with fewer systemic adverse reactions [22]. Considering that adenomyotic lesions are located deep within the pelvic cavity, rectal administration enhances drug absorption through the mucosal lining. Studies have shown that HM retention enemas alleviate pain in endometriosis models by reducing inflammation, peripheral sensitization, and pelvic adhesions [39], and are widely used in gynecological practice. Future studies are warranted to explore the clinical potential of diverse HM formulations for the treatment of endometriosis and related gynecological disorders.

Despite the therapeutic advantages of HM, the potential for unpredictable adverse events remains a concern, particularly when formulations involve multiple herbal components. A systematic review reported that the hepatobiliary system was the most frequently affected, followed by nervous and gastrointestinal systems. Commonly reported symptoms included nausea, diarrhea, and vomiting [40]. Another review found that the incidence of adverse events associated with HM varied widely, ranging from 0.03% to 29.84%, with a median pooled estimate of 1.42% [41]. However, in the present study, the incidence of adverse effects such as vaginal discharge, abdominal pain, vaginal bleeding, nausea, and vomiting was lower in the HM group when used as adjunct therapy following HIFU treatment for adenomyosis. The pooled analysis yielded a RR of 0.54, indicating a reduced risk of adverse events with HM co-administration. In addition, numerous previous studies have demonstrated the beneficial effects of HM in gynecological conditions and postoperative recovery. These include pain relief and wound healing after cesarean section [42], management of puerperal wind syndrome (Sanhupung) [43], and prevention of recurrence following endometrioma surgery [44]. Nonetheless, further research is warranted to investigate the potential contraindications of HM in gynecological diseases, particularly focusing on the identification of specific herbal components that may pose risks under certain physiological conditions such as pregnancy or lactation.

This study has several limitations. All 14 included RCTs were conducted in China, which may have introduced regional and publication biases. Additionally, variability in outcome measures among the studies contributed to heterogeneity, and the overall quality of evidence ranged from low to moderate. Although most of the included RCTs were from China, it is worth noting that research on HM and bioactive natural compounds is not limited to East Asia. A growing number of preclinical studies are being conducted in other countries, exploring the pharmacological mechanisms of herbal components. In recent years, the integration of nanotechnology with HM has emerged as a global research trend, with increasing contributions from countries such as the United States and India. This convergence offers novel opportunities to enhance the therapeutic efficacy of HM by addressing pharmacokinetic limitations such as poor solubility, low membrane permeability, rapid metabolism, and limited bioavailability. Incorporating HM-derived active compounds into nanocarrier-based delivery systems—including liposomes, solid lipid nanoparticles, and polymeric nanocarriers—has been shown to improve the solubility, stability, and tissue-specific delivery of herbal ingredients, thereby enhancing their bioavailability and therapeutic potential [45,46].

Our findings suggest the potential clinical utility of HM as a complementary strategy for managing recurrence and supporting recovery after HIFU therapy in patients with AM. This study may serve as a foundational reference for future research on the use of HM in postoperative care for AM. High-quality, rigorously designed RCTs, including those that assess diverse formulations and modes of administration, are needed to confirm the efficacy and safety of HM.

## 5. Conclusions

HM administration following HIFU therapy alleviates post-procedural adverse effects and significantly improves the symptoms of AM by reducing dysmenorrhea, menstrual volume, and serum CA125 and E2 levels. Although treatments such as LNG-IUS and GnRHa are commonly administered after HIFU therapy, HM may serve as a meaningful alternative for post-procedural management. Further clinical studies with long-term follow-up, including recurrence and pregnancy rates, should be conducted to expand the current evidence.

## Figures and Tables

**Figure 1 pharmaceuticals-18-00843-f001:**
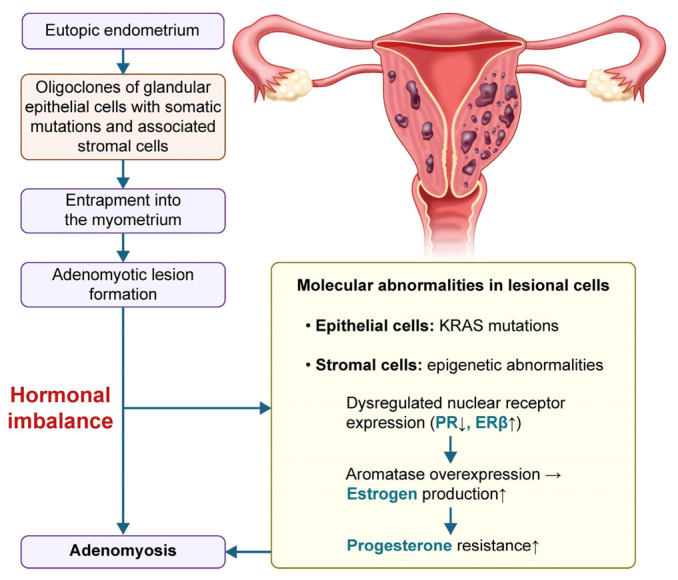
Hormone-driven pathogenesis of adenomyosis. KRAS, Kirsten rat sarcoma viral oncogene homolog; ERβ, estrogen receptor beta; PR, progesterone receptor.

**Figure 2 pharmaceuticals-18-00843-f002:**
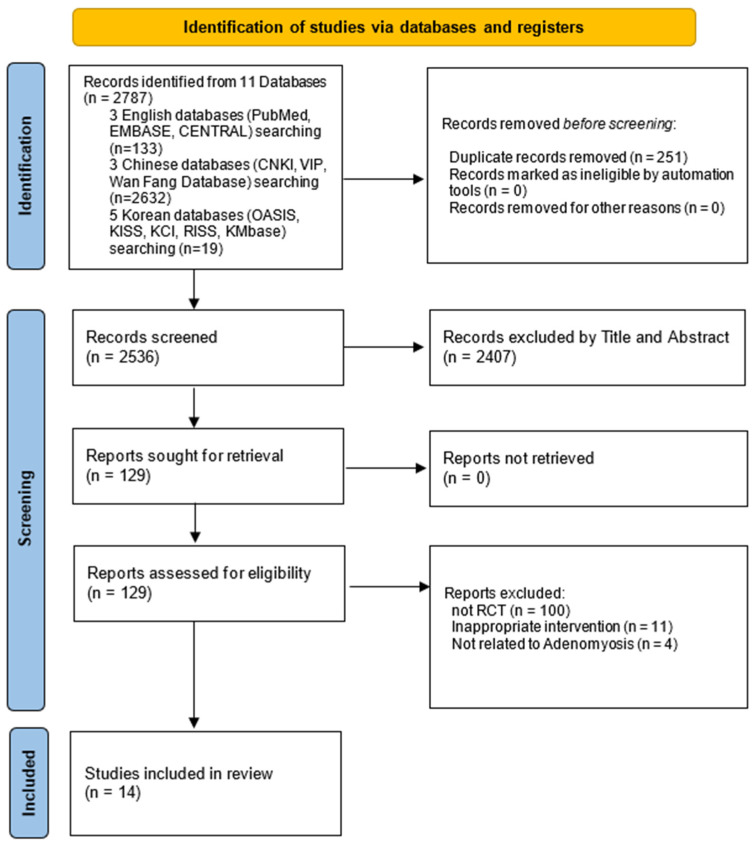
PRISMA flow diagram of study selection process. EMBASE, Excerpta Medica database; CENTRAL, Cochrane Central Register of Controlled Trials; CNKI, China National Knowledge Infrastructure; VIP, Chinese Scientific Journal Database; OASIS, Oriental Medicine Advanced Searching Integrated System; KISS, Korean Studies Information Service System; KCI, Korea Citation Index; RISS, Research Information Sharing Service; KMbase, Korean Medical Database.

**Figure 3 pharmaceuticals-18-00843-f003:**
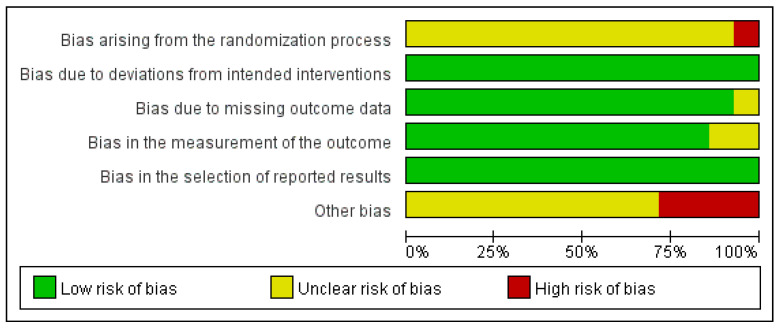
Risk of bias graph.

**Figure 4 pharmaceuticals-18-00843-f004:**
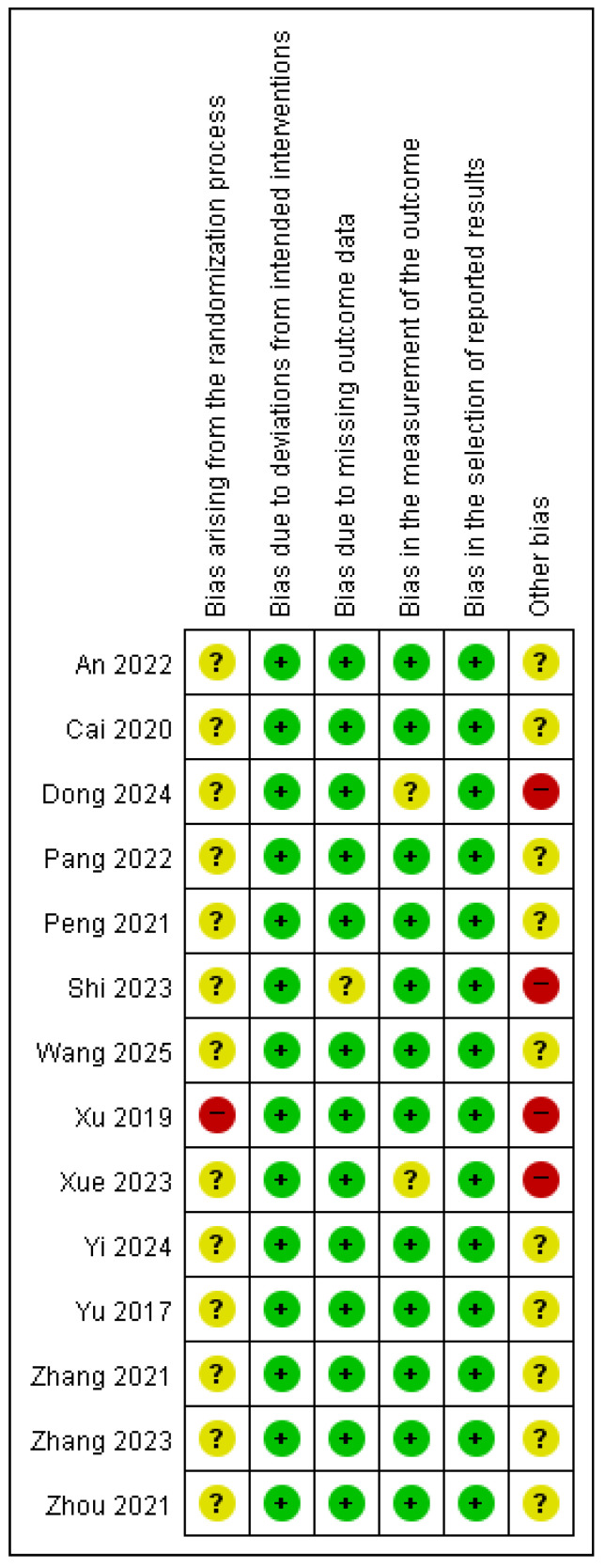
Risk of bias assessment [15,16,17,18,19,20,21,22,23,24,25,26,27,28]. +, Low risk of bias; ?, Unclear risk of bias; −, High risk of bias.

**Figure 5 pharmaceuticals-18-00843-f005:**
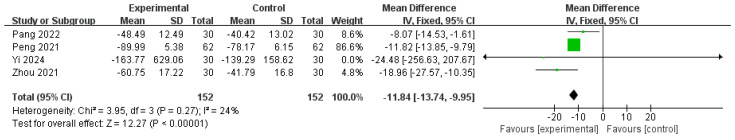
Forest plot of the uterine volume [18,19,24,28].

**Figure 6 pharmaceuticals-18-00843-f006:**
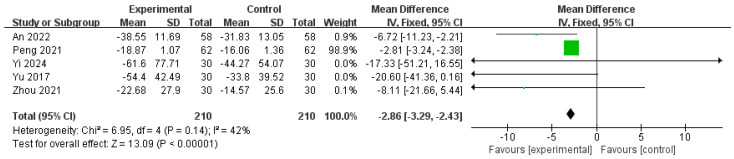
Forest plot of the adenomyotic lesion volume [15,19,24,25,28].

**Figure 7 pharmaceuticals-18-00843-f007:**
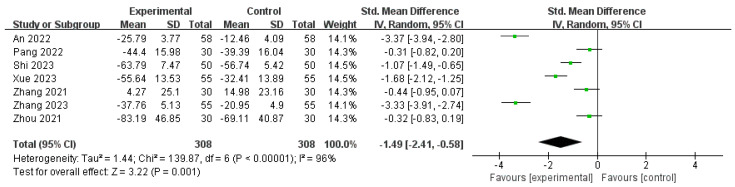
Forest plot of the CA125 levels [15,18,20,23,26,27,28].

**Figure 8 pharmaceuticals-18-00843-f008:**
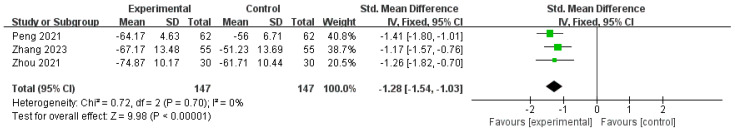
Forest plot of the E2 levels [19,27,28].

**Figure 9 pharmaceuticals-18-00843-f009:**
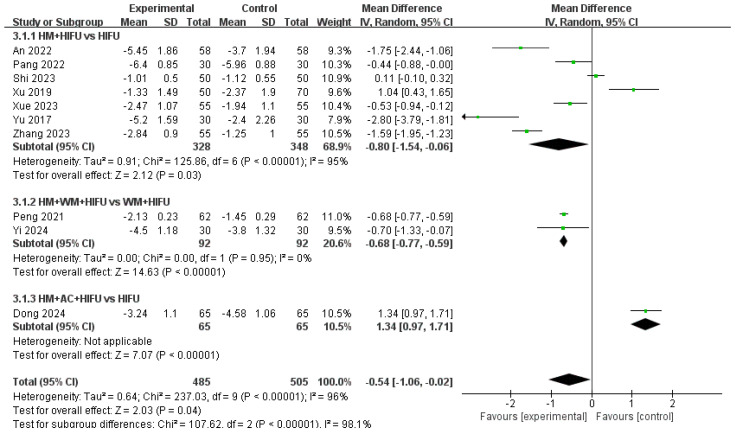
Forest plot of the VAS score for dysmenorrhea (Subgroup by intervention type) [15,17,18,19,20,22,23,24,25,27]. HIFU, High-intensity focused ultrasound; HM, herbal medicine; AC, acupucture; WM, western medicine.

**Figure 10 pharmaceuticals-18-00843-f010:**
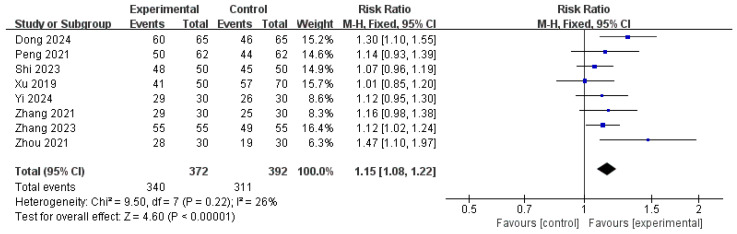
Forest plot of the total effective rate [17,19,20,22,24,26,27,28].

**Figure 11 pharmaceuticals-18-00843-f011:**
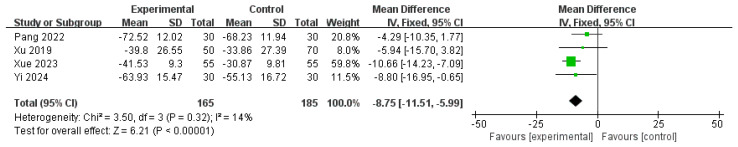
Forest plot of the PBAC score for menstrual volume [18,22,23,24].

**Figure 12 pharmaceuticals-18-00843-f012:**
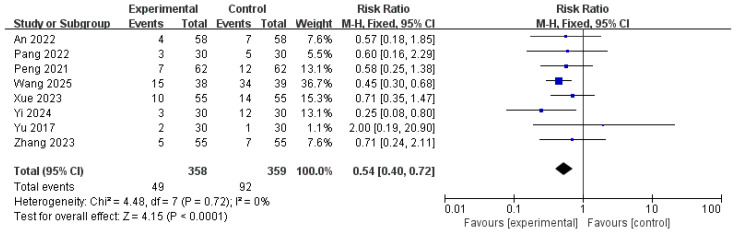
Forest plot of adverse events [15,18,19,21,23,24,25,27].

**Figure 13 pharmaceuticals-18-00843-f013:**
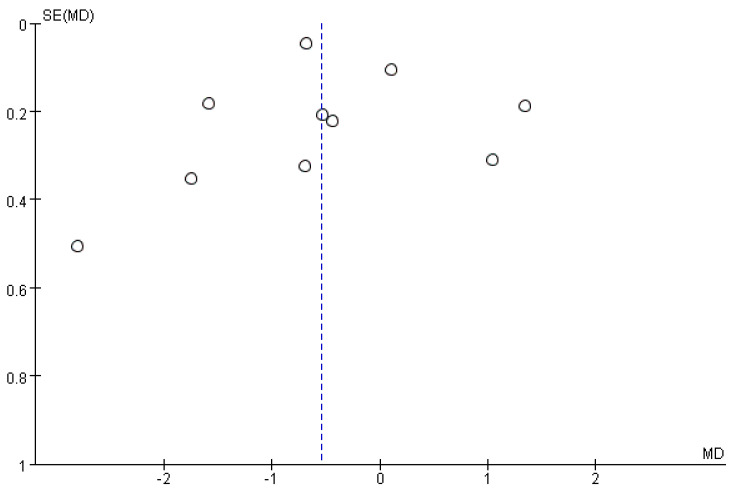
Funnel plot of the VAS score for dysmenorrhea.

**Table 1 pharmaceuticals-18-00843-t001:** Basic characteristics of the included studies.

First Author (Year)	Sample Size (E/C)	Age Distribution (yr, Mean ± SD)	Duration of Illness (yr, Mean ± SD)	Experimental Intervention (E)	Total Treatment Periods	Follow Up Periods	Outcome Measurement	Adverse Events (n/N)
				Control Intervention (C)				
An (2022) [15]	116 (58/58)	E: (34.72 ± 3.95) C: (35.26 ± 4.07)	E: (3.74 ± 0.85) C: (4.01 ± 0.96)	(C) + HM	3 m	6 m	(1)(2)(7)(8)(20)	E:4/58 C:7/58
				HIFU Treatment				
Cai (2020) [16]	66 (33/33)	E: (37.93 ± 4.82) C: (38.30 ± 4.31)	E: (5.67 ± 2.13) C: (5.92 ± 1.67)	(C) + HM	3 m	NR	(1)(2)(5)(8)	NR
				HIFU Treatment				
Dong (2024) [17]	130 (65/65)	E: (35.7 ± 12.8) C: (36.5 ± 13.1)	E: (4.5 ± 1.4) C: (5.3 ± 1.3)	(C) + HM + AC	3 m	NR	(1)(3)(6)(9)(13)	NR
				HIFU Treatment				
Pang (2022) [18]	60 (30/30)	E: (39.82 ± 2.75) C: (40.09 ± 2.72)	E: (2.16 ± 0.49) C: (2.21 ± 0.46)	(C) + HM	6 m	NR	(1)(2)(4)(5)(6)(10)(11)(20)	E:3/30 C:5/30
				HIFU Treatment				
Peng (2021) [19]	124 (62/62)	E: (35.02 ± 6.84) C: (34.87 ± 7.90)	E: (5.14 ± 1.05) C: (5.03 ± 0.83)	(C) + HM	3 m	6 m, 12 m	(1)(2)(3)(4)(6)(7)(20)(21)	E:7/62 C:12/62
				HIFU Treatment + LNG-IUS				
Shi (2023) [20]	100 (50/50)	E: (34.73 ± 6.24) C: (34.25 ± 6.83)	E: (5.52 ± 1.35) C: (5.25 ± 1.20)	(C) + HM	3 m	6 m	(1)(2)(3)(4)(5)(7)	NR
				HIFU Treatment				
Wang (2025) [21]	77 (38/39)	E: (40.71 ± 6.34) C: (40.18 ± 6.41)	NR	(C) + HM + AC	3 d	NR	(14)(15)(16)(17)(20)	E:15/38 C:34/39
				HIFU Treatment				
Xu (2019) [22]	120 (50/70)	E: (42.70 ± 5.04) C: (43.49 ± 4.42)	NR	(C) + HM	7 d	3 m, 6 m	(1)(3)(4)(9)(20)	NR
				HIFU Treatment				
Xue (2023) [23]	110 (55/55)	E: (37.89 ± 3.69) C: (38.46 ± 3.78)	E: (2.49 ± 0.69) C: (2.43 ± 0.65)	(C) + HM	3 m	NR	(1)(2)(4)(5)(6)(20)	E:10/55 C:14/55
				HIFU Treatment				
Yi (2024) [24]	66 (33/33)	E: (38.600 ± 4.073) C: (38.530 ± 4.216)	E: (224.667 ± 195.525) wk C: (331.367 ± 342.319) wk	(C) + HM	3 m	6 m	(1)(3)(4)(5)(6)(7)(12)(20)	E:3/30 C:12/30
				HIFU Treatment + GnRHa				
Yu (2017) [25]	60 (30/30)	E: (34.2 ± 2.86) C: (35.6 ± 3.12)	NR	(C) + HM	7 d	3 m	(1)(7)(20)	E:2/30 C:1/30
				HIFU Treatment				
Zhang (2021) [26]	60 (30/30)	E: (38.96 ± 1.13) C: (38.41 ± 1.90)	E: (5.87 ± 2.08) C: (6.02 ± 1.98)	(C) + HM	7 d	NR	(2)(3)(5)	NR
				HIFU Treatment + GnRHa				
Zhang (2023) [27]	110 (55/55)	E: (38.96 ± 1.13) C: (38.41 ± 1.90)	E: (2.11 ± 0.68) C: (2.09 ± 0.95)	(C) + HM	3 m	NR	(1)(2)(3)(5)(8)(18)(20)	E:5/55 C:7/55
				HIFU Treatment				
Zhou (2021) [28]	60 (30/30)	E: (43.47 ± 4.71) C: (42.98 ± 4.11)	E: (3.22 ± 0.78) C: (3.18 ± 0.74)	(C) + HM	3 m	6 m	(2)(3)(6)(7)(19)	NR
				HIFU Treatment				

E, experimental; C, control; SD, standard deviation; NR, not reported; d, day; wk, week; m, month; yr, year; HM, herbal medicine; HIFU, high-intensity focused ultrasound; AC, acupuncture; LNG-IUS, Levonorgestrel-Releasing Intrauterine System; GnRHa, Gonadotropin-Releasing Hormone agonist; (1), dysmenorrhea score; (2), blood test indicator; (3), total effective rate; (4), menstrual volume score; (5), tcm syndrome score; (6), uterine volume; (7), adenomyotic lesion volume; (8), menstrual volume; (9), endometrial thickness; (10), quality of life; (11), pregnancy rate; (12), cumulative incidence of low estrogen reaction; (13), chronic pelvic pain score; (14), operation condition; (15), intensity of pain during and post operation; (16), dosage of sedative and analgesic drugs; (17), postoperative houston pain outcome instrument; (18), menstrual cycle; (19), antral follicle count; (20), adverse events; (21), recurrence rate; NR, not reported.

**Table 2 pharmaceuticals-18-00843-t002:** Intervention details for the experimental group.

First Author (Year)	Type of Formulation/Composition	Dosage	Frequency (per Day)
An (2022) [15]	Decoction *Astragali radix* 15 g, *Poria sclerotium* 10 g, *Cinnamomi ramulus* 10 g, *Paeoniae radix* 10 g, *Moutan radicis cortex* 10 g, *Persicae semen* 6 g, *Sparganii rhizoma* 6 g, *Curcumae rhizoma* 6 g	200 mL	2 times
Cai (2020) [16]	Aifu Nuangong Decoction *Astragali radix*, *Leonuri herba* 20 g, *Cyperi rhizoma*, *Cnidii rhizoma*, *Paeoniae radix alba*, *Morindae radix*, *Corydalis tuber*, *Linderae radix*, *Typhae pollen* 15 g, *Angelicae gigantis radix*, *Angelicae dahuricae radix*, *Artemisiae argyi folium*, *Dipsaci radix*, *Cinnamomi ramulus* 10 g	150 mL	2 times
	External application *Astragali radix*, *Cyperi rhizoma*, *Cnidii rhizoma*, *Morindae radix*, *Corydalis tuber*, *Linderae radix*, *Typhae pollen*, *Artemisiae argyi folium*, *Dipsaci radix*, *Cinnamomi ramulus* 10 g, *Paeoniae radix alba*, *Leonuri herba*, *Angelicae gigantis radix*, *Angelicae dahuricae radix* 15 g	4~6 h	1 time
Dong (2024) [17]	Bushen Huoxue Decoction *Hominis placenta*, *Cuscutae semen*, *Ligustri fructus*, *Angelicae gigantis radix*, *Cnidii rhizoma*, *Paeoniae radix* 15 g, *Olibanum*, *Myrrha* 10 g	200 mL	2 times
	Acupuncture Sameumgyo(SP6)(Bilateral), Gwanwon(CV4), Jigi(SP8), Sinsu(BL23), Yoyanggwan(GV3)(Unilateral)	30 min	1 time
Pang (2022) [18]	Nei Yi Pill * *Draconis sanguis*, *Salviae miltiorrhizae radix*, *Angelicae gigantis radix*, *Notoginseng radix et rhizoma*, *Persicae semen*, *Curcumae rhizoma*, *Sparganii rhizoma*, *Cyperi rhizoma*, *Cinnamomi ramulus*, *Achyranthis radix*	10 g	2 times
Peng (2021) [19]	Danggui Shaoyao San Pill *Paeoniae radix alba* 20 g, *Angelicae gigantis radix* 10 g, *Atractylodis rhizoma alba* 10 g, *Cnidii rhizoma* 10 g, *Alismatis rhizoma* 20 g, *Poria sclerotium* 30 g	1 pill	3 times
Shi (2023) [20]	Bushen Huoxue Decoction *Cuscutae semen* 30 g, *Cistanchis herba*, *Poria sclerotium*, *Lycii fructus*, *Eucommiae cortex*, *Rehmanniae radix preparata* 20 g, *Curcumae rhizoma*, *Sparganii rhizoma*, *Myrrha*, *Angelicae gigantis radix* 15 g, *Carthami flos* 10 g Menstrual period: *Corni fructus*, *Psoraleae semen* 20 g Non-Menstrual period: *Typhae pollen* 20 g, *Crataegi fructus*, *Trogopterorum faeces* 15 g, *Notoginseng radix et rhizoma* 10 g	NR	2 times
Wang (2025) [21]	Jiawei Sini Sijunzi Decotion *Bupleuri radix* 10 g, *Aurantii fructus immaturus* 10 g, *Paeoniae radix alba* 10 g, *Glycyrrhizae radix et rhizoma Praeparata cum melle* 5 g, *Codonopsis pilosulae radix* 10 g, *Poria sclerotium* 10 g, *Atractylodis rhizoma alba praeparata* 10 g, *Dioscoreae rhizoma* 20 g, *Massa medicata fermentata* 15 g, *Corydalis tuber* 10 g	150 mL	3 times
	Acupuncture and Auricular acupuncture Shenmen(TF4), Sympathetic autonomic(AH6a), Sameumgyo(SP6)(Bilateral)	NR	NR
Xu (2019) [22]	External application *Angelicae gigantis radix* 10 g, *Paeoniae radix* 10 g, *Artemisiae argyi folium* 30 g, *Acanthopanacis cortex* 15 g, *Osterici seu notopterygii radix et rhizoma* 15 g, *Speranskiae tuberculatae herba* 15 g, *Angelicae dahuricae radix* 10 g, *Foeniculi fructus* 30 g, *Spatholobi caulis* 15 g, *Carthami flos* 15 g, *Araliae continentalis radix* 15 g	NR	1 time
Xue (2023) [23]	Bushen Huoxue Sanyu Decoction *Euonymi ramuli suberalatum*, *Corydalis tuber*, *Paeoniae radix*, *Cnidii rhizoma*, *Angelicae gigantis radix*, *Hominis placenta*, *Cuscutae semen* 10 g, *Citri unshius pericarpium immaturus* 6 g	200 mL	2 times
Yi (2024) [24]	Enema: Wenjing Tongzu Decoction *Cinnamomi ramulus*, *Paeoniae radix*, *Linderae radix*, *Sparganii rhizoma*, *Curcumae rhizoma*, *Cyperi rhizoma*, *Corydalis tuber*, *Artemisiae anomalae herba*, *Lycopi herba*, *Cnidi fructus*, *Spatholobi caulis*, Cynanchi paniculati radix et rhizoma, *Speranskiae tuberculatae herba* 30 g, *Ephedrae herba*, *Poria sclerotium*, *Morindae radix* 20 g, *Hirudo*, *Asiasari radix et rhizoma*, *Olibanum*, *Myrrha* 15 g	100 mL	1 time
Yu (2017) [25]	Enema *Ilicis pubescentis radix*, *Lonicerae folium et caulis*, *Rhei radix et rhizoma*, *Prunellae spica*, *Gleditsiae spina*, *Liquidambaris fructus*, *Sparganii rhizoma*, *Curcumae rhizom*, *Salviae miltiorrhizae radix*, et al.	100 mL	1 time
Zhang (2021) [26]	Decoction *Rhei radix et rhizoma* 3 g, *Cnidii rhizoma* 9 g, *Lonicerae flos* 9 g, *Trogopterorum faeces* 12 g, *Angelicae gigantis radix* 15 g, *Scrophulariae radix* 15 g, *Taraxaci herba* 15 g, *Typhae pollen* 15 g, *Sargentodoxae caulis* 18 g, *Paeoniae radix alba* 30 g, *Patriniae radix* 30 g, *Notoginseng radix et rhizoma* 3 g	NR	1 time
Zhang (2023) [27]	Granules *Rehmanniae radix preparata* 15 g, *Codonopsis pilosulae radix* 15 g, *Sparganii rhizoma* 10 g, *Curcumae rhizoma* 10 g, *Olibanum praeparata cum aceto* 10 g, *myrrha praeparata cum aceto* 10 g, *Liquidambaris fructus* 15 g, *Ephedrae herba* 6 g, *Cinnamomi cortex* 10 g, *Hirudo praeparata* 3 g, *Crataegi fructus* 15 g	NR	2 times
Zhou (2021) [28]	Epimedii Herba Mixture Decoction *Epimedii herba* 15 g, *Cistanchis herba* 15 g, *Polygoni multiflori radix praeparata* 10 g, *Cuscutae semen* 15 g, *Codonopsis pilosulae radix* 15 g, *Astragali radix* 20 g, *Curcumae rhizoma* 15 g, *Salviae miltiorrhizae radix* 15 g, *Paeoniae radix* 15 g, *Corydalis tuber* 15 g, *Meliae fructus* 15 g, *Persicae semen* 10 g, *Olibanum* 10 g, *Myrrha* 10 g, *Achyranthis radix* 10 g	NR	2 times

NR, not reported; min, minutes; h, hour; SP, Spleen Meridian; CV, Conception Vessel; BL, Bladder Meridian; GV, Governing Vessel; TF, Triangular Fossa; AH, Auricular Helix; *, The study did not provide detailed information regarding the specific dosage administered.

**Table 3 pharmaceuticals-18-00843-t003:** The quality of evidence.

Outcomes	No. Participants (Studies)	Anticipated Absolute Effects (95% CI)		Relative Effect (95% CI)	Heterogeneity (I^2^)	Quality of Evidence (GRADE)
		Risk with Control Group	Risk with Intervention Group			
Uterine volume	304 (4 RCTs)	-	MD 11.84 lower (13.74 lower to 9.95 lower)	-	24	⨁⨁◯◯ Low ^a,b^
Adenomyotic lesion volume	420 (5 RCTs)	-	MD 2.86 lower (3.29 lower to 2.43 lower)	-	42	⨁⨁◯◯ Low ^a,b^
CA125	616 (7 RCTs)	-	SMD 1.49 lower (2.41 lower to 0.58 lower)	-	96	⨁⨁◯◯ Low ^a,b,c^
E2	294 (3 RCTs)	-	SMD 1.28 lower (1.54 lower to 1.03 lower)	-	0	⨁⨁◯◯ Low ^a,b^
VAS scores of dysmenorrhea	990 (10 RCTs)	-	MD 1.51 lower (2.02 lower to 1 lower)	-	96	⨁⨁⨁◯ Moderate ^c,d^
Total effective rate	764 (8 RCTs)	793 per 1000	119 more per 1000 (63 more to 175 more)	RR 1.15 (1.08 to 1.22)	26	⨁⨁⨁◯ Moderate ^d^
PBAC score for menstrual volume	350 (4 RCTs)	-	MD 8.75 lower (11.51 lower to 5.99 lower)	-	14	⨁⨁◯◯ Low ^b,d^
Adverse effect	717 (8 RCTs)	256 per 1000	118 fewer per 1000 (154 fewer to 72 fewer)	RR 0.54 (0.40 to 0.72)	0	⨁⨁⨁◯ Moderate ^a^

^a^, The overall bias is unclear in half or more of the studies; ^b^, The sample size did not meet the OIS criterion; ^c^, although heterogeneity was substantial (I^2^ > 70%), all studies showed effect estimates in the same direction, and the variability in effect sizes was considered clinically acceptable. Therefore, no downgrade was applied for inconsistency; ^d^, the overall risk of bias was unclear in most of the included studies, with one study assessed as having a high risk of bias; CI, confidence interval; RR, risk ratio; MD, mean difference; SMD, standardized mean difference; RCT, randomized controlled trial; GRADE, Grading of Recommendations Assessment, Development, and Evaluation; CA, cancer antigen; E2, estradiol; VAS, visual analog scale; PBAC, pictorial blood assessment chart; OIS, optimal information size; ⊕, represents higher certainty of evidence; ◯, represents lower certainty of evidence.

## Data Availability

All data analyzed in this study are included in this published article and Appendix A.

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
