# Peer review of "The Efficacy and Safety Herbal Medicine for Symptom Management After HIFU Treatment in Adenomyosis: A Systematic Review and Meta-Analysis"

_pharmaceuticals, 2025, doi:10.3390/ph18060843_

Round 1

Reviewer 1 Report

Comments and Suggestions for Authors

Authors have presented an interesting systemic analysis entitled The Efficacy and Safety Herbal Medicine for Symptom Management After HIFU Treatment in Adenomyosis: A systematic review and meta-analysis, which can be further considered addressing the below given suggestions. 

  1. Does it is necessary to include authors name abbreviation and degree after title of manuscript in author detail section?, if not suggested to remove.
  2. Authors are suggested to add a graphical illustration showing the Adenomyosis (AM) is a hormone-dependent gynecological condition.
  3. The search strategy showing data collected till February (line no. 21) and later review protocol was registered in March at INPLASY, why is it so?
  4. Line no. 129, 141: a third reviewer (S.Y.M.); Two independent reviewers (E.J.K. and Y.S.S.), correct to “author” in place of reviewer.
  5. Suggested to include a subsection in section 2.4 on exclusion criteria, is process of screening includes some specific key words in title or in abstract, or did the authors read each and everything of the paper to confirm the suitability in the review.
  6. Authors are encouraged to include the name of specific bioactive compounds for the plants name indicated in the discussion section, to support the management strategy of HM.
  7. Since the author reported the major limitation Is major papers reported after conducting study on Chinese, how the future of HM research may progress.
  8. Moreover, authors should indicated how the HM are considered potent in the management of Adenomyosis, compared to synthetic drugs which can be guided via incorporating them in novel drug delivery based carriers.  

Author Response

We have diligently reviewed the comments provided by the reviewers and have carefully incorporated revisions. We have uploaded the final revised version incorporating the reviewer's comments and detailed the modifications in the attached file which are marked in red. We hope the corrections will meet with your approval.

Reviewer 2 Report

Comments and Suggestions for Authors

Adenomyosis is a benign disease of the uterus characterized by the presence of endometrial tissue (the mucosa that lines the uterus) within the muscular wall of the uterus (myometrium). Treatment options for adenomyosis vary depending on the severity of the condition and the age of the woman. Treatment strategies range from drug therapy to surgical treatment. Among the less invasive treatments, high-intensity focused ultrasound (HIFU) therapy is used, a non-invasive technique that uses ultrasound, inducing an increase in temperature in the pathological tissues until their complete destruction. Currently, herbal medicine, or phytotherapy, is also used to treat various pathologies, a branch of pharmacotherapy that deals with the prevention and treatment of various disorders and diseases through the use of medicinal plants and preparations obtained from them. The present study aims to perform an updated and methodologically rigorous systematic review and meta-analysis to evaluate the efficacy and safety of herbal medicine, as an adjuvant to HIFU therapy for the treatment of adenomyosis.

The topic covered is of great interest given the symptoms and implications that this uterine pathology can determine.

Overall, despite some weaknesses (also highlighted by the authors themselves), the review has an adequate structure and summarizes numerous data on the aspects covered.

However, some modifications are necessary, as reported below.

Herbal medicine is a focal aspect of this work, therefore, the authors should include a short paragraph on its use for the treatment of diseases, but also on possible contraindications. In fact, herbal medicine, despite its reputation as a natural and safe cure, may have contraindications and side effects similar to those of traditional drugs, such as Allergic reactions, Interactions with other drugs or foods, Pregnancy and breastfeeding, Specific contraindications, Side effects, Toxicity, Interactions with surgical interventions.

The captions of Tables and Figures must be revised to report their description in a concise but clear way without having to resort to the text.

Author Response

(The authors gave the same response as above.)

Round 2

Reviewer 1 Report

Comments and Suggestions for Authors

The authors have reflected all the said suggestions and comments, which made the manuscript enhanced with improved readability; Thus, I suggest for further consideration with acceptance.